# The Coverage and Acceptance Spectrum of COVID-19 Vaccines among Healthcare Professionals in Western Tanzania: What Can We Learn from This Pandemic?

**DOI:** 10.3390/vaccines10091429

**Published:** 2022-08-30

**Authors:** Eveline T. Konje, Namanya Basinda, Anthony Kapesa, Stella Mugassa, Helmut A. Nyawale, Mariam M. Mirambo, Nyambura Moremi, Domenica Morona, Stephen E. Mshana

**Affiliations:** 1Department of Biostatistics and Epidemiology, School of Public Health, Catholic University of Health and Allied Sciences–BUGANDO, Mwanza P.O. Box 1464, Tanzania; 2Department of Community Medicine, School of Public Health, Catholic University of Health and Allied Sciences–BUGANDO, Mwanza P.O. Box 1464, Tanzania; 3Department of Microbiology and Immunology, Weill Bugando School of Medicine, Catholic University of Health and Allied Sciences–BUGANDO, Mwanza P.O. Box 1464, Tanzania; 4National Public Health Laboratory, Dar es Salaam P.O. Box 9083, Tanzania; 5Department of Parasitology, Weill Bugando School of Medicine, Catholic University of Health and Allied Sciences–BUGANDO, Mwanza P.O. Box 1464, Tanzania

**Keywords:** vaccines, acceptance rate, health professionals, hesitancy, Tanzania

## Abstract

The vaccination rate against COVID-19 remains low in developing countries due to vaccine hesitancy. Vaccine hesitancy is a public health threat in curbing COVID-19 pandemic globally. Healthcare professionals have been found to play a critical role in vaccine advocacy and promotion campaigns in the general population. A cross sectional study was conducted in the initial months of the COVID-19 vaccination roll out program in Tanzania to determine the acceptance rate, perceived barriers, and cues for actions. A total of 811 healthcare professionals participated from 26 health facilities in western Tanzania. The World Health Organization (WHO) vaccine acceptance questionnaire was adopted with minor modifications to capture the local contexts and used in data collection. Only (18.5%) healthcare professionals had received a COVID-19 vaccine and acceptance rate was 29%. The majority (62%) of participants were in the hesitancy stage due to issues related to lack of effective communication and reliable information regarding efficacy and safety. In this era of COVID-19 pandemic, there is a need to engage and involve public health figures and opinion leaders through transparent dialogue to clarify vaccine-related safety, quality, and efficacy. These strategies will reduce misconception, mistrust, and improve uptake among healthcare professionals and eventually in the general population.

## 1. Introduction

Globally, the number of confirmed coronavirus disease of 2019 (COVID-19) cases is reported to be over 536 million and over 6.3 million deaths as of 19 June 2022 [1]. The Africa region has reported 9.0 million cumulative COVID-19 cases and 173,000 deaths [1]. Vaccines have been used as public health measure to break the chain of transmission and mutation for the purpose of curbing infectious diseases [2]. The available COVID-19 vaccines remain a sustainable solution to prevent morbidity and mortality and complement other public health measures such as face mask-wearing, social distancing, hand washing, and hand sanitizing [3]. Vaccine hesitancy and refusal continue to threaten universal coverage of COVID-19 vaccination in low- and middle-income countries (LMICs) [2,4,5]. It is estimated that at least 60–70% of the population should be vaccinated to curb transmission and reach herd immunity [3,6]. To achieve this, the World Health Organization (WHO) has outlined the actions required by global community to vaccinate 40% of the world population against COVID-19 by the end of 2021, and 70% by June 2022 [3].

Despite the COVID-19 Vaccines Global Access (COVAX) initiative program that aimed at facilitating equitable access and distribution of COVID-19 vaccine globally, the coverage is still low among those who are eligible for vaccination in the countries that have received vaccines [7,8]. Over 11.3 billion doses of COVID-19 vaccines have been administered worldwide but only 11% of the population in low-income countries are vaccinated, compared to 73% of those in high-income countries [8]. As of 27 June 2022, African region had received 833.5 million doses and only 18% of the populations were fully vaccinated [8]. Vaccines remain the key public health preventive measure with sustainable effect in preventing, containing, and stopping transmission of SARS-CoV-2 globally [3]. Nevertheless, uptake for the COVID-19 vaccine in the population is lagging behind which may slow down the global effort in combating the pandemic [7,8].

In a review by Wang et al., the pooled potential acceptance rate was reported to be 65.6% among health professionals with a higher rate (82%) displayed in the general population [9]. During outbreak, health professionals can be a vehicle for transmission and they are most likely to contract the SARS-CoV-2 infection during the COVID-19 pandemic. Health professionals have been classified as a higher risk group and have been given priority in receiving COVID-19 vaccination since they are the backbone of the health system during pandemics [3,8]. Recent studies on the willingness for COVID-19 vaccine uptake have reported high hesitancy to the COVID-19 vaccine due to safety, quality, and trust issues among health professionals [9].

Vaccine hesitancy refers to the unwillingness or refusal to receive a vaccine despite availability and accessibility of vaccination services [10]. Existing cultural, social, and political issues might have influenced the response to the COVID-19 pandemic across African countries [11]. Tanzania is among the countries that delayed joining the COVAX initiative program after opting for local context COVID-19 preventive measures and advocacy of the COVID-19 vaccine efficacy evaluation before its distribution might have impacted health beliefs and risks assessment of different populations [12]. In view of this, the study was designed to shed light on the overall COVID-19 vaccine acceptance spectrum in the initial months of the COVID-19 vaccine rollout program and coverage among the health professionals in Mwanza, northwestern Tanzania. By doing so, we can learn how to promote and improve the vaccine coverage in the general population.

## 2. Materials and Methods

A cross sectional study was conducted in September 2021 from selected public health facilities in Mwanza region. The study involved health professionals from five out of seven districts, namely Ilemela, Nyamagana, Misungwi, Magu, and Ukerewe. The Mwanza region has a total of ~400 health facilities with 2500 health care providers of different levels of training. Health professionals from 23 health facilities in five districts who were present during data collection period participated in this study.

Using a potential acceptance rate of 50% from the review that reported a range of 43–94% [9], we required a total of 384 participants based on Leslie Kish’s formula [13]. To capture existing variation across different levels of health facilities, an effect size of two was used that led to sample size of 768. In this study, we recruited 811 health professionals who were available during the study period. The participants were distributed as follows: dispensary (48, 5.9%), health center (184, 22.7%), district hospital (214, 26.4%), regional hospital (132, 16.3%), and tertiary hospital (233, 28.7%) (Appendix A).

A convenience sample of health professionals from different cadres (i.e., nurses, clinical officers, medical officers, and specialists) who were available at the health facility during data collection period (between 13 and 26 September 2021) responded to the structured questions. No sampling technique was applied during selection of participants since we aimed at involving all health professionals who were present at work during the data collection period. A self-administered approach was used to allow privacy when going through the vaccination acceptance scale.

The data collection procedure was conducted within the health facility premises at different units/departments to ensure that all health professionals were reached. The distribution of questionnaires was done by medical students from the Catholic University of Health and Allied Sciences (CUHAS). We used a structured questionnaire that was adopted from the WHO vaccine acceptance scale to determine the perceived level of acceptance, hesitancy, and refusal among health professionals.

Questions were constructed to capture reasons for not accepting the vaccine from different perspectives, namely safety and quality of vaccine, efficacy, conspiracy, trust, and perception on signing consent forms. The acceptance spectrum consisting of five categories (full acceptance, low hesitancy, high hesitancy, undecided, and refusal) was centered on the following question: “A vaccine to protect against COVID-19 is available in Tanzania, would you get vaccinated as soon as possible?” Five different responses were used to categorize participants. (1) The full acceptance group includes all participants who reported receipt of vaccination or definitely agreed to receive vaccine; (2) the low level hesitancy group consists of those who responded with possibly yes; (3) the high level hesitancy group involves participants who responded with probably not; (4) the undecided group involves participants who were not sure; and (5) the refusal group involves participants who said definitely not to the vaccine. The acceptance rate was calculated by dividing the total number of participants who reported to be vaccinated or those who were willing to receive COVID-19 vaccine with total participants (i.e., 811).

Data entry was done using MS Excel and transferred to STATA version 13 for analysis. Simple descriptive statistics were performed to obtain the key characteristics of study participants. Graphs were used to display acceptance spectrum levels and reported reasons for not accepting the COVID-19 vaccine. During analysis, we merged participants in refusal, undecided, low, and high levels of hesitancy to formulate one group (hesitancy) to compare with those who accepted the vaccine. The vaccine coverage, acceptance rates, and hesitancy levels were reported with 95% confidence intervals. A Chi-square test was used to compare categorical variables. Logistic regression was performed to determine the factors associated with hesitancy among healthcare professionals. A *p*-value of less than 0.05 was considered statistically significant for all analyses.

This study was conducted according to the Declaration of Helsinki. It was approved by the joint Catholic University of Health and Allied Sciences/Bugando Medical Centre (CUHAS/BMC) research ethics and review committee (CREC) with ethical clearance number CRECU/2181/2021. Before commencing the study, permission was sought from regional and district administrative authorities. The participants were informed of the aim of the study and provided with a self-administered and structured questionnaire to fill in. Confidentiality was observed and maintained throughout the study period.

## 3. Results

### 3.1. Demographic Characteristics and General Health Behavior of Participants

This study involved health care professionals from clinical units/departments at different levels of public health facilities. More than two-thirds of participants (579, 71%) were from hospitals (i.e., district, regional, and tertiary hospitals). The average age of participants was 35 ± 9.04 years and males made up slightly more than half of the sample (423, 52%). Most of the participants (419, 52%) reported to have a certificate or diploma level of medical or nursing training followed by those with a degree and Master’s (287, 35%). The median years working in the medical field was five with interquartile range (IQR) of 2 to 10 years. General health condition and healthy behavior were as follows: 113 (14%) reported having chronic disease, only 47 (6%) reported smoking cigarettes, more than half (465, 57%) reported doing physical exercise regularly, and 444 (55%) received the Hepatitis B vaccine. Furthermore, proportions of hesitancy were significantly high in health centers and hospitals, in lower age groups, and among nurses/doctors/specialists. See Table 1 below.

### 3.2. COVID-19 Vaccine Status among Health Professionals in Mwanza Western Tanzania

The overall COVID-19 vaccine uptake was only 18.5% (95% CI: 16.0%–21.3%) among health professionals from the surveyed sites. The uptake level across the districts was almost the same for Ukerewe, Magu, and Mwanza city whereas Misungwi district displayed a relatively lower status than other districts (see Figure 1). No statistical difference (*p*-value > 0.05) was observed in COVID-19 uptake for Mwanza city, Magu, and Ukerewe. However, there was a significant difference when comparing COVID-19 vaccine uptake between Misungwi district and other districts; Mwanza city (*p*-value = 0.039), Magu (*p*-value = 0.039), and Ukerewe (*p*-value = 0.017).

In Figure 2, we compare the uptake of COVID-19 and Hepatitis B vaccines, which are both available to health professionals. We found that there was a significant difference (*p*-value < 0.05) in uptake of COVID-19 and Hepatitis B vaccines in each site. Figure 2 shows that more than half of participants reported having received the Hepatitis B vaccine compared to those who received COVID-19 vaccine. However, not all health professionals received the Hepatitis B vaccine.

### 3.3. COVID-19 Vaccine Acceptance Level among Health Professionals in Western Tanzania

The vaccine acceptance spectrum ranges from definitely accept to definitely refuse (Figure 3). In this study, less than one third (232, 29%) of all health professionals fully accepted the COVID-19 vaccine of which two-thirds (150, 65%) reported being vaccinated at the time of survey. In Figure 3 below, it can be observed that the majority (63%) of health professionals who participated in this study were in delayed stage. This stage consists of health professionals from low hesitancy to undecided stage due to different reasons. Furthermore, a small group of ~8% health professionals were in the refusal stage.

### 3.4. Perceived Barriers for Not Taking the COVID-19 Vaccine among Health Professionals

Several perceived barriers of not taking the COVID-19 vaccine were related to conspiracy, efficacy of the vaccine, attitude towards signing of consent form, safety of the vaccine, and trust. These barriers could explain the avoidance or hesitancy behavior towards COVID-19 vaccine uptake, which is of public health importance. In Figure 4, more than three-quarters of all participants (749, 80%) were concerned about safety and whether the vaccine is trustworthy. A higher proportion (517, 69.4%) of health professionals who were hesitant perceived that issues related to safety for the COVID-19 vaccine hindered its uptake compared to those who accepted (232, 53.8%) with *p*-value < 0.05. The perceived issues on lack of reliable information during pandemic and about COVID-19 vaccine, elapsed short time for vaccine development, and fear of unknown side effects interfered with participants’ decision on its uptake. Another concern was related to signing the consent form as a way of taking individual responsibility for unknown consequences of vaccine. More than one-third (749, 39%) of participants reported signing of the consent form as a barrier for vaccine uptake. However, this issue was highly mentioned by those who accepted the vaccine compared to those who hesitated to take up the vaccine (232, 46.9%) vs. (517, 38.1%) with a *p*-value = 0.022. Only a few health professionals mentioned conspiracy issues hindering the uptake of the COVID-19 vaccine, which was not statistically significant between hesitancy and acceptance groups. See Figure 4.

### 3.5. Factors Associated with Hesitancy of COVID-19 Vaccine among Health Professionals

In Table 2, the following factors were significantly associated with hesitancy of the COVID-19 vaccine with *p*-value < 0.05 at univariate analysis: age below 50 years, reported not having chronic disease (hypertension, diabetes mellitus, asthma etc.), no receipt of Hepatitis B vaccination, and working at the hospital level. At multivariate analysis, only age below 50 years and not having received Hepatitis B vaccine were significantly associated with COVID-19 vaccine hesitancy. Thus, participants aged 30–49 years were two times more likely to hesitate over the COVID-19 vaccine and those below 30 years were three times more likely to hesitate receiving COVID-19 vaccine compared to participants aged 50 years and above. For those reporting receipt of Hepatitis B vaccination, they were two times more likely to hesitate receiving COVID-19 vaccination (aOR = 2.19, 95% C.I. 1.56–3.06) compared to their counterpart. In both univariate and multivariate analyses, there was no association between COVID-19 vaccine hesitancy and sex of a participant.

### 3.6. Cues for Actions on Improving COVID-19 Vaccine Uptake among Health Professionals

The key cues that were supported by almost half of health professionals include availability and provision of information, social support, and involvement of influential leaders during the advocacy campaign to improve COVID-19 vaccine uptake among health professionals. A majority of participants reported that engagement of government authority for the provision of vaccine information, involvement of public figures in advocacy of the vaccine, and support from close family members and friends would improve the vaccine’s uptake. Declaring the COVID-19 vaccine to be mandatory to all health professionals was supported by only a third, leaving a majority either undecided or disagreeing with that notion (Figure 5).

## 4. Discussion

Vaccine hesitancy is becoming a public health concern for curbing the COVID-19 pandemic. In this study, we found that the vaccine coverage was low among health professionals. Two-thirds of health professionals were in the hesitancy stage due to issues related to safety; followed by one-third of participants because of consent form signing; and a few mentioned about efficacy of vaccine and conspiracy. Furthermore, the majority of participants perceived that lack of effective communication and paucity of reliable information hinder COVID-19 vaccination services and coverage.

As of 31 September 2021, less than a quarter of health professionals reported receipt of the vaccination despite the availability and accessibility of the services in Mwanza, western Tanzania. In this study, the COVID-19 vaccine coverage rate was lower (18.5%) than the coverage rates reported in different countries; Sierra Leone (38%) [14], Ethiopia (62%) [15], and Malawi (83%) [16]. In high-income countries, almost universal vaccine coverage was documented among healthcare workers and more than 70% of the population are fully vaccinated [1,8]. In the current study, the acceptance rate of the COVID-19 vaccine was only 29% with almost two-thirds of health professionals being in a delay phase (hesitancy phases). This is relatively similar to the acceptance rate reported in Egypt (21–30%) [17]. Our study has reported the lowest potential acceptance rate among health professionals compared to other studies in LMICs such as Nepal (38%), Nigeria (44%), Ethiopia (48%), India (63%), and Uganda (70%) [9,17,18]. A pooled COVID-19 vaccine acceptance was estimated to be less than a half (48%) among healthcare workers in Africa which could suggest low uptake among general population [3,8,19,20]. In European countries, the acceptance rate was reported to be above 75%, although it tends to change over time [21,22,23,24,25,26]. For instance, in Germany, the overall acceptance rate was 92% among health workers with 49% of them being vaccinated [25]. A similar situation was reported in Canada with more than 80% acceptance rate among health workers [24]. The low vaccination uptake in our setting could be explained by different factors related to social, cultural, and political reasons rather than knowledge inadequacy or lack of human and financial resources [27,28,29,30,31]. Most countries in Africa reported fewer COVID-19 cases and mortality compared to European countries, which might have influenced the uptake of vaccination services [1]. Although the low burden of COVID-19 in Africa region could be associated with poor health system and detection capacity, it was the region least affected by the pandemic [1]. The initial predictions suggested a worst situation in LMICs which was not the case; with this, health professionals might have considered themselves to be at low risk of COVID-19. Furthermore, a country like Tanzania took a different approach in curbing the COVID-19 with emphasis on public health preventive measures such as hand-washing, social distancing, divine intervention, and use of local remedies for steaming inhalation referred to as “*nyungu*” or “*kujifukiza*” [32]. Initially, COVID-19 was not accepted as a public health problem by the authority and the community was being assured to be safe without vaccine [32]. This approach could have negatively affected the decision for the uptake of the vaccination services among health professionals when the pandemic was acknowledged by the Tanzania Government. Furthermore, high uptake of COVID-19 vaccination in some of developed countries could be explained partly by introduction of a mandatory vaccination order and a higher number of cases and mortality in the region. In developed countries such as France, German, Italy, Greece, and England, to mention a few, they have made the COVID-19 vaccine mandatory for social and healthcare workers [33,34,35]. Although, the introduction of a mandatory vaccination order has received different opinions of whether it is ethically justifiable or not [36]. Hence, the local advocacy strategies can be explored and implemented to improve COVID-19 vaccination coverage in LMICs. Additionally, the low acceptance level among health professionals may cause a great challenge on national and global efforts to curb the COVID-19 pandemic in LMICs [3,8]. It is well documented that health professionals have a great role in advocacy for immunization program in the community [37,38]. For instance, Smith et al. reported that parents/guardians who received advice from health professionals on health-related issues including the safety of vaccines were more likely to vaccinate their children [37]. The hesitancy among health professionals in our setting and other LMICs may pose a public health threat to combat the transmission of COVID-19 and emergence of new strains.

The perceived barriers for uptake of the new vaccine were related to safety and trust, efficacy, and conspiracy, which may occur due to different reasons. In this study, similar barriers were reported contributing to the low uptake of the COVID-19 vaccine among health workers, which were also documented across different countries globally [5,9,14,21,39]. A well-informed community of nurses and clinicians would be expected to show a high level of understanding on the importance of vaccination as a way of containing infectious disease during the COVID-19 era. However, health professionals displayed doubts on the general quality and efficacy of the new vaccine, suggesting a lack of clear involvement and engagement in open dialogue between scientists and end users [9,28,39]. The assumption that scientists know and the community consumes might has led to a dark period among end users of the vaccine [27,28]. The hesitancy level among health professionals might suggest a missed opportunity in community engagement and health education for combating COVID-19 in the population.

Achieving universal coverage for COVID-19 vaccine at a population level remains to be a promising strategy in combating transmission and reemergence of new strains of SARS-CoV-2 [3,8]. Currently, the world is in another wave with some countries being highly affected by a new phase of COVID-19 [1]. As of 6 July 2022, confirmed COVID-19 cases were around 548 million and 63% of people were fully vaccinated globally but only 24% were fully vaccinated in Africa [40]. The cues for action include involvement of health public figures and opinion leaders in promoting and mobilizing the community starting with provision of reliable information to health professionals on vaccine safety, importance of preventing new infections or the reemergence of new SARS-CoV-2 strains [27,29,31,41]. In this study, information was not sufficient to allow people to make informed decisions as shown by most of participants remaining undecided on whether the use of vaccine will be appropriate or not. Furthermore, use of social media as source of health information in the era of COVID-19 proved to increase anxiety and resistance on uptake of vaccination services [42].

In this study, young healthcare professionals (<50 years) were identified to be more likely to hesitate in the receipt of the COVID-19 vaccination as it has been documented elsewhere [14,15,19,24,43]. This could be because of reported low risk of coronavirus infection and disease severity in this population [44,45] as well as exposure to misinformation on vaccine safety and efficacy that was consumed by young population via social media [46]. We also observed low uptake of the Hepatitis B virus vaccine that is associated with COVID-19 vaccination hesitancy among healthcare workers, underscoring the importance of advocating routine vaccines. In additional, it was observed that the Hepatitis B vaccine coverage was only half among health professionals who received at least one dose of Hepatitis B virus vaccine. Previous research on coverage of Hepatitis B virus among healthcare workers reported a range of 30% to 78% with a median of 50% [47]. In East African countries, the overall Hepatitis B infection rate has been reported to be 6% with other countries reporting as high as 26% [14,48,49]. Studies have documented the high burden of the Hepatitis B virus among healthcare workers, although the uptake of complete Hepatitis B virus vaccination remains low in most countries [14,48,49]. This may suggest the need for an advocacy campaign to emphasize the importance of routine vaccines among healthcare workers.

Most studies determined a willingness to vaccinate prior to the availability and accessibility of COVID-19 vaccine across the countries [4,7,17,18,19]. However, we determined the coverage of the COVID-19 vaccine among a priority population who might have influenced the trust of the community soon after the rollout program in Tanzania. Furthermore, this was a cross sectional survey which provided information or highlighted the situation in a snapshot. The observed situation might have changed dramatically depending on the availability of information from reliable sources, the level of the individual risk assessment, and other drivers that influence the decision of health professionals. All these drivers can change the acceptance level either negatively or positively. Furthermore, the survey utilized a convenience sample which may lead to selection bias, since only those who were willing and present during data collection period were included, hence the participants may have not represented all other healthcare professionals who were not involved in this study.

## 5. Conclusions

In conclusion, uptake of the COVID-19 vaccine among health professionals was low, with less than a quarter being vaccinated across all surveyed districts in western Tanzania as of September 2021. Although acceptance level was almost a third, there is still hope from those on the delayed stage to be vaccinated when appropriate measures will be considered to facilitate their decision. Lack of communication and reliable information on the safety, efficacy, side effects, and taking individual responsibility on unknown consequences when signing consent forms were reported to hinder the uptake of the COVID-19 vaccine. The key strategies to improve uptake of COVID-19 vaccine among health professionals may include: (1) involvement of trusted professionals and public health figures to allay fears through transparent dialogue/interactive information sessions that respects and addresses health care workers’ concerns on safety and efficacy of COVID-19 vaccine, (2) respective authority to communicate in a timely manner addressing uncertainty with clarity on safety and efficacy of the COVID-19 vaccine, (3) engagement of influential figures in dialogue to highlight the benefit of the vaccine to the global community (4) health facility management to advocate the uptake of routine vaccines periodically, and (5) lastly, the government to cooperate with young community champions, religious leaders, respected figures, or organizations to advocate safety and efficacy of vaccine.

## Figures and Tables

**Figure 1 vaccines-10-01429-f001:**
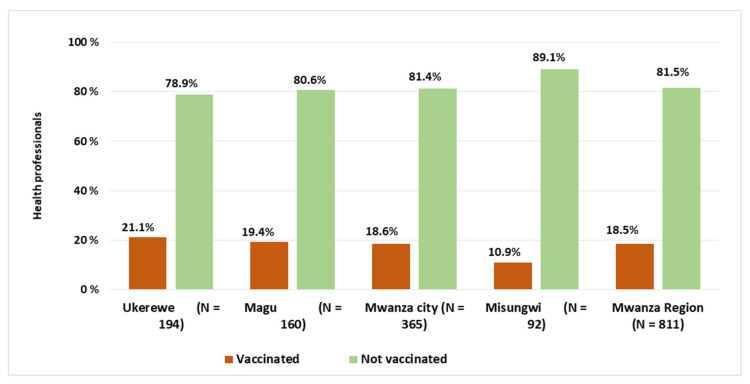
COVID-19 vaccine status among health professionals.

**Figure 2 vaccines-10-01429-f002:**
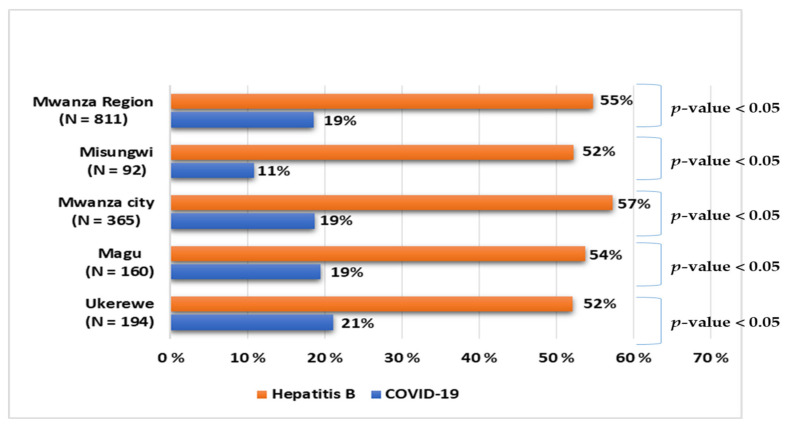
Comparison of Hepatitis B and COVID-19 vaccine status among health professionals.

**Figure 3 vaccines-10-01429-f003:**
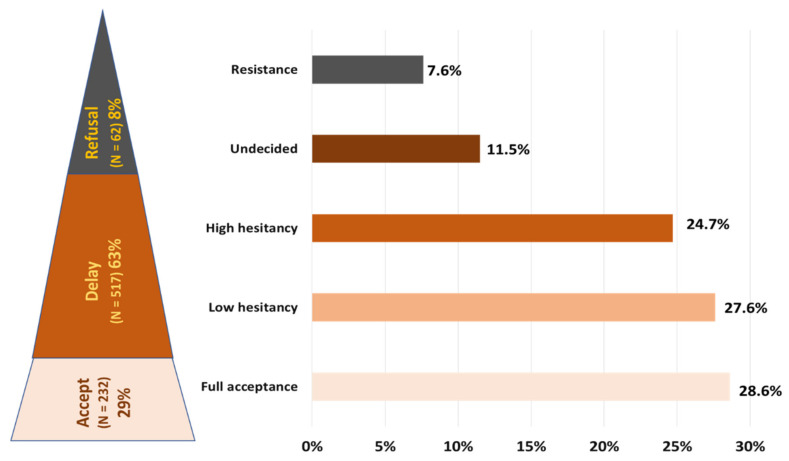
COVID-19 vaccine acceptance spectrum among health professionals.

**Figure 4 vaccines-10-01429-f004:**
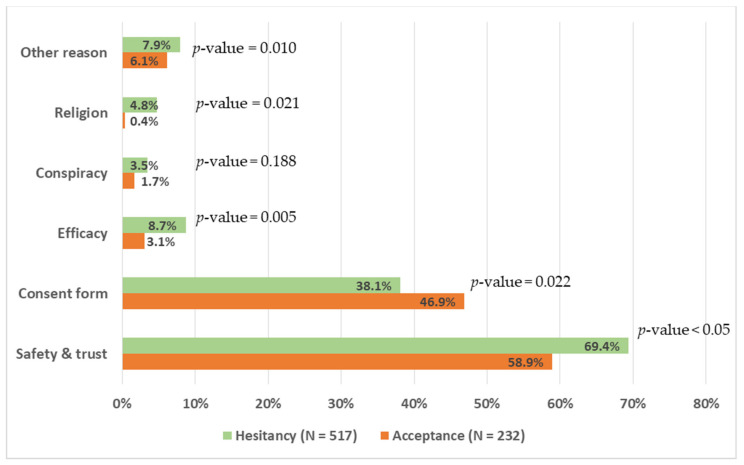
Perceived barriers for not accepting COVID-19 vaccine by acceptance level.

**Figure 5 vaccines-10-01429-f005:**
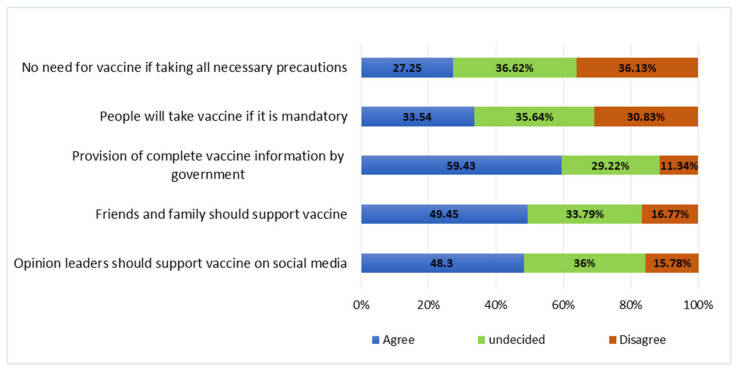
Cues for action to work on improving the COVID-19 vaccine uptake among health professionals.

**Table 1 vaccines-10-01429-t001:** General characteristics and reported health behaviors of health professionals.

Characteristic	N (%)	Acceptance, N = 323 N (%)	Hesitancy, N = 579 N (%)	Chi^2^ *p*-Value
Level of health facility				
Tertiary hospital	233 (28.7)	63 (27.2)	170 (29.4)	0.001
Regional hospital	132 (16.3)	20 (8.6)	112 (19.3)	
District hospital	214 (26.4)	66 (28.5)	148 (25.6)	
Health center	184 (22.7)	64 (27.6)	120 (20.7)	
Dispensary	48 (5.9)	19 (8.2)	29 (5.0)	
Age group				
Less than 30 years	295 (36.4)	60 (25.9)	235 (40.6)	0.000
30–49 years	446 (55.0)	135 (58.1)	311 (53.7)	
50 years and above	70 (8.6)	37 (16.0)	33 (5.7)	
Gender				
Female	388 (47.8)	119 (51.3)	269 (46.5)	0.213
Male	423 (52.2)	113 (48.7)	310 (53.5)	
Education level				
Primary	51 (6.3)	22 (9.5)	29 (5.0)	0.011
Secondary	54 (6.7)	22 (9.5)	32 (5.5)	
Certificate/diploma	419 (51.7)	107 (46.1)	312 (53.9)	
Degree/Master’s/PhD	287 (35.4)	81 (34.9)	206 (35.6)	
Cadre				
Medical attendant	105 (12.9)	44 (18.9)	61 (10.5)	0.004
Nurse/clinical officer	419 (51.7)	107 (46.1)	312 (53.9)	
Doctor/specialist	287 (35.4)	81 (34.9)	206 (35.6)	
Smoking status (yes)	47 (5.8)	15 (6.5)	32 (5.5)	0.605
Exercise regularly (yes)	465 (57.3)	133 (57.3)	332 (57.3)	0.997
Chronic conditions (yes)	113 (13.9)	45 (19.40)	68 (11.7)	0.004
Name of conditions				
Hypertension	51 (45.1)			
Diabetes mellitus	24 (21.2)			
Asthma	17 (15.0)			
Heart disease	7 (6.2)			
Cancer	4 (3.5)			
Others	10 (8.9)			

**Table 2 vaccines-10-01429-t002:** Factors associated with COVID-19 vaccine hesitancy (univariate and multivariate analyses).

Variable	Univariate Analysis			Multivariate Analysis		
	OR	95% C.I.	*p* Value	aOR	95% C.I.	*p*-Value
Sex						
Female	1					
Male	1.21	0.90–1.66	0.196			
Age group						
≥50 years	1					
30–49 years	2.58	1.55–4.31	0.000	2.20	1.28–3.76	0.004
<30 years	4.39	2.54–7.60	0.000	3.31	1.83–5.98	0.000
Education level						
Degree and above	1					
Certificate/Diploma	1.15	0.82–1.61	0.428	1.45	0.99–2.12	0.055
Primary/Secondary	0.55	0.34–0.87	0.011	0.86	0.49–1.51	0.595
Presence of chronic Disease						
Yes	1			1		
No	1.81	1.19–2.73	0.005	1.29	0.83–2.02	0.251
Uptake of Hepatitis B vaccine						
Yes	1			1		
No	2.25	1.63–3.09	0.000	2.19	1.56–3.06	0.000
Facility Level						
Dispensary	1					
Health Center	1.23	0.64–2.36	0.537	0.97	0.48–1.96	0.938
Hospital	1.89	1.03–3.47	0.040	1.66	0.83–3.31	0.151

## Data Availability

The dataset that was used this study is available upon request to the corresponding author.

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
