# Peer review of "The Coverage and Acceptance Spectrum of COVID-19 Vaccines among Healthcare Professionals in Western Tanzania: What Can We Learn from This Pandemic?"

_vaccines, 2022, doi:10.3390/vaccines10091429_

Round 1

Reviewer 1 Report

I’m thankful to review the paper entitled “The coverage and acceptance spectrum of COVID 19 vaccines among healthcare professionals in Western Tanzania: What can we learn from this pandemic” for Vaccines MDPI Journal.

My decision is Accept after major revision.

The paper is interesting and deal with the emerging topic such a Vaccines hesitancy. The introduction is clear and well arranged -could be improved. The methodology sounds good but the statistical analyses mast be improved. The discussion is good even could be improved.

Introduction

·        Please insert to introduction-section studies about the vaccine acceptance in pre-endemic period such a: Mfinanga SG, Mnyambwa NP, Minja DT, Ntinginya NE, Ngadaya E, Makani J, Makubi AN. Lancet. 2021 Apr 24;397(10284):1542-1543. doi: 10.1016/S0140-6736(21)00678-4. Epub 2021 Apr 14.

Materials and methods

·         Please add the acceptance rate of participants

Enhance the discussion with the follow references:

·        COVID-19 Vaccine Hesitancy among Healthcare Workers and Trainees in Freetown, Sierra Leone: A Cross-Sectional Study. Yendewa SA, Ghazzawi M, James PB, Smith M, Massaquoi SP, Babawo LS, Deen GF, Russell JBW, Samai M, Sahr F, Lakoh S, Salata RA, Yendewa GA. Vaccines (Basel). 2022 May 11;10(5):757. doi: 10.3390/vaccines10050757.

        Suggested also to the authors to discussed the low proportion of Hepatitis-B Vaccine because the study concerns health professionals

Results  

·        Suggested to authors to reform from Table 2 to Table 1 and insert the vaccine coverage/acceptance according to health profession.

·        Suggested to authors in statistical analysis to investigate the possibility to compare   in two tables analysis as one variable the participants with high level education against the participants with low level education and coverage/acceptance of vaccination. 

Limitations: please add limitations for the study in separate paragraph (make it clear).

References

Follow the instructions of the journal, all references must be reform:

·        References must be numbered in order of appearance in the text (including table captions and figure legends) and listed individually at the end of the manuscript. We recommend preparing the references with a bibliography software package, such as EndNote, Reference Manager or Zotero to avoid typing mistakes and duplicated references. We encourage citations to data, computer code and other citable research material. If available online, you may use reference style 9. below.

·        Citations and References in Supplementary files are permitted provided that they also appear in the main text and in the reference list.

In the text, reference numbers should be placed in square brackets [ ], and placed before the punctuation; for example [1], [1–3] or [1,3]. For embedded citations in the text with pagination, use both parentheses and brackets to indicate the reference number and page numbers; for example [5] (p. 10). or [6] (pp. 101–105).

Author Response

Thank you for your constructive comments that have improved our manuscript. I have attached a detailed response below.

Reviewer 2 Report

The authors present the results of a survey regarding vaccine hesitancy in Western Tanzania during the early phases of coronavirus vaccine rollouts.  Their findings are of interest due to the low uptake among a key population of healthcare workers.  Their work is well presented but several areas could be improved and strengthen their findings and conclusions.

My first concern begins in results section 3.4 (line 169).  It is not clear which group or groups were analyzed.  Based on their conclusions, they correctly state that individuals in the delayed hesitancy group are the appropriate target for addressing hesitancy.  Was this analysis performed to look at differences between those that refuse the vaccine, those with delayed hesitancy, and those who accepted the vaccine?  A comparison of the factors between these groups would further focus the conclusions and direct interventions to the target group.

My second concern is regarding the statistical approach.  The authors only provide descriptive statistics without performing any statistical analysis evaluating significant differences.  The authors should seek out statistical help if needed and perform simple statistical analyses (chi-square, others as appropriate) to strengthen their conclusions.  These are the two reasons why I state the methods must be improved.

Several minor concerns:

Figures/graphs should include numbers of respondents, not just percentages where appropriate.

Table 2 – significant digits in the percentage column should be limited to 1 decimal point, same for Figure 4 and 5.  Figure 3, the x-axis should not have decimal points.

It would be interesting for the discussion to include a comparison to reports from Europe or North America.

Line 220-222 – Figure 4 partially addresses this comment with the question about not believing the vaccine is necessary.  The authors may want to comment further.

The authors include IRB determination at the end of the manuscript.  I usually see a brief statement in the methods section as well, or reference to the IRB/consent determination at the end of the manuscript.

Lines 96 and 276 – convenient should be convenience sample or “survey utilized convenience sampling”

Supplemental table - please subtotal the districts, do the authors know the total number of eligible participants in each district or is there potential for significant bias? 

Careful proofreading is required as there are numerous grammatical mistakes throughout the manuscript.

Author Response

(The authors gave the same response as above.)

Round 2

Reviewer 1 Report

Accept in present form.

Author Response

Thank you for your valuable inputs that improved our manuscripts

Reviewer 2 Report

The authors present their revised manuscript evaluating vaccine hesitancy in Tanzania.  The inclusion of statistical methods greatly strengthens the manuscript and informs their conclusions.  I have some remaining concerns the author’s should incorporate.

First, the elegant statistical analysis is not utilized to its strengths in the discussion.  Line 247 states conspiracy and efficacy were concerns for hesitant health care professionals, although this was not a statistically supported conclusion.  Importantly, the authors found strong associations with age and receipt of Hepatitis B vaccination.  Both of these findings have been reported elsewhere and are not surprising.  The authors could better discuss the importance of this finding relative to strategies to improve vaccine uptake in Tanzania. In particular, strategies addressing safety among young health care personnel would provide the greatest improvements in vaccine uptake.

Second, they discuss the profound lack of uptake, not only in Tanzania but other LMICs, especially relative to Germany and Canada.  They would benefit from more in depth discussion to answer: Why this difference?  What needs to be done to address this disparity?  These would be new avenues of investigation opened by their original research.

Minor issues:

Numerous grammatical errors persist, COVID19 is used interchangeably with COVID-19.  Line 39 – coronavirus should be one word.  Line 43, “The available COVID-19 vaccines remain a sustainable…”  etc.  Careful proofreading is necessary as there are multiple additional errors.

Figure 1 and 2, please provide the numbers of respondents and statistical analysis – the differences are obvious but performing the statistics would help.  Second, are the significant differences based on geography?  Misungwi has lower uptake.

Figure 4 should show the differences based on the breakdown between groups as now described nicely in the results section.

Author Response

Dear Reviewer,

We appreciate for your constructive inputs that shaped our manuscript dramatically. 
